# Risk Factors for Surgical Wound Infection and Fascial Dehiscence After Open Gynecologic Oncologic Surgery: A Retrospective Cohort Study

**DOI:** 10.3390/cancers16244157

**Published:** 2024-12-13

**Authors:** Carolin Hagedorn, Nadja Dornhöfer, Bahriye Aktas, Laura Weydandt, Massimiliano Lia

**Affiliations:** Department of Gynecology, University Hospital of Leipzig, 04103 Leipzig, Germany; nadja.dornhoefer@medizin.uni-leipzig.de (N.D.); bahriye.aktas@medizin.uni-leipzig.de (B.A.); laura.weydandt@medizin.uni-leipzig.de (L.W.); massimiliano.lia@medizin.uni-leipzig.de (M.L.)

**Keywords:** surgical side infection, fascial dehiscence, gynecological oncology, risk factors, open surgery

## Abstract

This research addresses the critical issue of surgical site infections (SSI) and fascial dehiscence (FD) in patients with gynecological cancer, a group often overlooked in existing studies. We aim to identify key risk factors associated with these complications following open surgery by focusing on a specific patient population. We showed that the effect of the duration of surgery depends on whether bowel surgery was performed or not. Similarly, the effect of BMI on the occurrence of SSI is influenced by the patient’s age, with younger patients experiencing a significantly steeper rise in their risk of SSI with increasing BMI. The findings could significantly impact the medical community by highlighting the need for tailored prevention strategies, particularly when bowel surgery is involved or in younger patients with obesity. This research aims to improve patient outcomes and enhance the overall quality of care for those undergoing surgery for gynecological malignancies.

## 1. Introduction

Surgical side infections (SSI) [1] and fascial dehiscence (FD, also referred as burst abdomen or wound dehiscence) are potentially serious postoperative complications that increase morbidity and necessitate medical intervention [2]. FD requires immediate surgical treatment, prolongs hospital stays and increases healthcare costs, thus emphasizing its clinical importance [3,4,5,6,7].

Accurate identification of patients at high risk of these wound complications is important, as interventions to prevent them have been proposed [8,9,10,11,12,13,14]. Several studies have described risk factors and developed multivariable prediction models for these outcomes. However, these prediction models were usually developed on very heterogeneous cohorts [15], and women with gynecological malignancies were often excluded [16]. Additionally, those studies focusing on patients with gynecological malignancies failed to address model discrimination by means of the areas under the curve (AUC). Importantly, current prediction models do not account for statistical interactions (i.e., how certain factors modify the effect of another factor), which can have a relevant impact on the accuracy of a prediction model [17,18,19]. Consequently, existing models may not be sufficiently accurate in predicting SSI or FD after open surgery for gynecological malignancies.

In this study, we investigated whether known risk factors are applicable in women with gynecological cancer (cervical, endometrial or ovarian cancer). Furthermore, we systematically searched for interaction effects between variables and included these into multivariable regressions. Hereby, we aimed to provide further evidence to describe the interplay between various risk factors for wound complications and thus to improve prediction.

## 2. Methods

This was a retrospective single-center cohort study including all patients with suspected gynecological cancer, who underwent median laparotomy at the Department of Gynecology, University of Leipzig, Germany, from January 2017 to December 2020. This research was approved by the Ethics Committee of the medical faculty of the University of Leipzig (007/24-ek; date: 30 January 2024). All patients were treated by at least one surgeon with gynecologic oncology subspecialty. Patients with immunosuppressant therapy, undergoing laparoscopic or robotic surgery, or primary vacuum-assisted closure (i.e., application of vacuum-assisted closure (VAC)) were excluded. Furthermore, pregnant patients and those in whom histological examination revealed benign disease were excluded.

The primary outcomes of interest were postoperative SSI and FD occurring within 30 days after the operation [1]. SSI was defined as any clinical diagnosis of wound infection, degradation of the continuity of the skin and subcutaneous tissue (with intact abdominal fascia) or the need for VAC therapy [1]. FD was defined as a total degradation of the fascial continuity requiring surgical intervention.

We collected data on all potential predictor variables linked to wound complications (SSI and FD): histologic type of the malignant disease (cervical, endometrial or ovarian cancer) [16,17,18,20], body mass index (BMI) [7,19], classification of physical status (American Society of Anesthesiologists (ASA) score) [7,14], age [7,21], diabetes mellitus [21,22], smoking [3,21], preoperative hemoglobin [23] and total serum protein levels [13,14,19], duration of the operation [3,14,19], bowel surgery (intended or accidental opening of the intestinal lumen) [17,24], extension of laparotomy over the umbilicus [13], neoadjuvant chemotherapy [8], and crystalloids administered during surgery [25]. To improve interpretation, patients’ age and BMI were scaled per 10 years and units, respectively. In accordance with this journal’s guidelines, we will provide our data for independent analysis by a selected team for additional data analysis or for the reproducibility of this study in other centers if requested.

### 2.1. Standard of Care

During the study period, laxatives were used prior to surgery at our institution. Intraoperative administration of antibiotics (cefuroxime and metronidazole or clindamycin in case of penicillin allergy) was routine, and prophylaxis was repeated if the blood loss exceeded one liter or the duration of surgery exceeded twice the half-life of the antibiotics used. The abdominal fascia was closed using the Smead–Jones technique [26] (Ethicon^®^ PDS™ II 1 loop) and skin closure was performed with skin staplers (Covidien Appose™ ULC skin staplers) or subcuticular suture (Ethicon^®^ monocryl 2-0). Postoperatively, early enteral nutrition was implemented if possible. Intra-abdominal drainages were removed at the latest by the third postoperative day unless decided otherwise by the surgeon due to the intraoperative situation (i.e., surgery of the bowel, spleen or bladder).

### 2.2. Statistical Analyses

Potential predictor variables were initially screened using univariable logistic regression to identify possible associations with the two primary outcomes. All variables with a *p*-value of <0.25 were included in the multivariable regression analysis to assess which variables were independent predictors in the second step [27]. We systematically searched for possible interaction variables (i.e., the effect of a parameter on the outcome depends on another parameter) in the following systematic way: First, variables significantly associated with the primary outcome were included in an algorithm to generate all possible interaction terms between these variables. The algorithm then used both, the variables and their interaction terms, to create all possible models. Second, these models were evaluated using the Akaike Information Criterion (AIC) to determine their predictive ability for the primary outcome (SSI or FD). The 100 best-performing models were examined for the interaction terms they contained. Lastly, the interaction terms present in these well-performing models were included in the multivariable model if the model-averaged importance of the interaction term was >0.8 [28].

To avoid multicollinearity, all continuous variables were centered by subtracting the arithmetic mean from each value.

For data analysis and graphical editing, we used the statistical software R (Version 4.3.2.). The “rms” package (Version 6.8-1) was used for multivariable logistic regression, “glmulti” (Version 1.0.8) was used to determine relevant statistical interactions, and “ggplot2” (Version 3.5.1) was used for graphics design.

## 3. Results

Between January 2017 and December 2020, a total of 204 women with gynecological malignancies were treated via median laparotomy at our institution. Of these patients, 99 (48.5%) had cervical cancer, 75 (36.8%) ovarian cancer and 30 (14.7%) suffered from endometrial cancer. SSI occurred in 50 (24.5%) and FD in 18 (8.8%) patients—patients with FD were also counted as SSI. The incidence of SSI and FD by tumor entity is shown in Figure 1D. Other patient characteristics are listed in Table 1.

Univariable screening identified the following parameters to be associated (*p* < 0.25) with SSI, which were consequently included into the multivariable model: woman’s age and BMI, diabetes, ASA score, extension of laparotomy over the umbilicus, skin closure technique, type of cancer, preoperative hemoglobin levels, number of crystalloids given during the operation, bowel surgery and operation time. Screening for interaction terms revealed two potentially important: between age and BMI and between operation time and bowel surgery. Specifically, the effect of operation time on the rate of SSI changed significantly depending on whether any bowel surgery was performed (Figure 1A). Similarly, the effect of increasing BMI on SSI was significantly diminished with increasing age (Figure 1C).

The significant variables listed above were included in the multivariable model (Table 2). Variables maintaining statistical significance were bowel surgery (*p* = 0.024) and BMI (*p* = 0.028). Operation time had a significant independent effect only in women who underwent bowel surgery (*p* = 0.039). Specifically, in the absence of bowel surgery, the adjusted odds ratio (OR) for each additional hour of operation time was 0.98. However, if bowel surgery was performed, the adjusted OR for each additional hour was 1.49 (0.98 multiplied by the interaction term 1.52).

Additionally, the effect of BMI was significantly decreased (*p* = 0.013) with increasing age. Specifically, the adjusted OR of BMI had to be multiplied by 0.48 (interaction term) for each additional decade of age.

Univariable screening determined the following parameters as associated (*p* < 0.25) with FD: diabetes, women’s age, type of cancer, ASA score, the number of crystalloids administered during operation, bowel surgery, and operation time. Screening for interactions suggested a potentially relevant interaction between bowel surgery and operation time. This interaction and its effect on FD are graphically presented in Figure 1B.

The significant variables mentioned above were included into the multivariable model (Table 3). The only variable maintaining statistical significance was the interaction term between operation time and bowel surgery (*p* = 0.007). Consequently, the only independent variable predicting FD was operation time if bowel surgery was part of the intervention. Conversely, operation time was not associated with a higher rate of FD if bowel surgery was not performed (*p* = 0.06). Therefore, bowel surgery per se was not associated with FD, but it had a significant effect through operation time.

Table 2 and Table 3: Models resulting from univariable and multivariable regression. In continuous variables, the odds ratio (OR) represents the change in the odds per unit shown in brackets. As an example, the odds of SSI increase by an OR of 2.51 in a patient with a BMI of 35 kg/m^2^ compared to a patient with a BMI of 25 kg/m^2^ (increase of 10 kg/m^2^ in the multivariable model). Statistical interaction terms (asterisk) represent the change in odds if a certain variable is present. As an example, the odds of SSI are unaffected by the duration of the operation (adjusted OR of 0.98 per hour). However, if bowel surgery is present, an additional adjusted OR of 1.52 per every additional hour of operation time has to be counted in.

## 4. Discussion

### 4.1. Summary of the Main Results

The main finding of this study is the association between extended operation time in open surgery for gynecologic malignancies and the risk of both SSI and FD. However, this effect was only significant if bowel surgery was part of the surgical procedure. Conversely, if bowel surgery was not performed, the risk of SSI or FD was not significantly influenced by the duration of surgery. Furthermore, our data suggest that the effect of BMI on the rate of SSI depends on the women’s age. Specifically, younger patients showed a significantly steeper increase in their risk of developing SSI with increasing BMI. To the best of our knowledge, this is the first study analyzing and demonstrating the presence of interactions between risk factors for SSI and FD in patients with gynecological cancer.

### 4.2. Results in the Context of the Published Literature

Various researchers have observed the effect that surgical duration is associated with both SSI [15,19] and FD [16]. However, operation times have usually been arbitrarily dichotomized in various studies [15], thus ignoring the fact that the risk of wound complications increases continuously [17] with increased surgery duration. We confirmed this fact in our cohort of patients with gynecological cancer (Figure 1A,B). Furthermore, statistical interactions (in which the effect of one predictor depends on that of another) are often not explored, although this is recommended when developing predictive models [29]. Research focusing on risk factors for SSI and FD in gynecological oncological patients mostly ignores statistical interactions [17,18], making predictions potentially less accurate. Our results indicate a possible interaction between bowel surgery and operation duration—we found that extended operation time combined with bowel surgery was strongly associated with the occurrence of SSI and FD in our cohort. Further studies are needed to examine its significance.

Patients’ age played a relevant role in the rate of SSI, although it did not show an independent significant effect in our cohort. However, age indirectly affected SSI risk by modifying the effect of BM. Specifically, an increased BMI had a stronger effect on SSI in younger patients than in older patients (Figure 1C). In summary, BMI was an independent predictor of SSI after open surgery for gynecological malignancies, but the magnitude of this effect depended on the women’s age.

The incidence of surgical site infections in our cohort was notably higher at 24.5% compared to the 2–5% range typically reported in previous studies on gynecological operations. However, these studies mainly involved cases of benign conditions [3,6,7,9]. Nevertheless, our results are consistent with the general incidence of SSI in Germany (24%) [22] and more comparable with the incidence of SSI in gynecological malignancies (approximately 10–34%) [17,18,19,30].

Previous studies on this topic identified several risk factors for wound complications in different cohorts of patients [2,5,6,7,10,21]. However, gynecologic patients are often underrepresented or completely excluded in the cohorts on which these studies are based [16]. Importantly, surgeries for gynecological malignancies are affected by specific bacterial contaminations, as hysterectomy leads to contact with the vaginal environment and the intra-abdominal region [3,23]. Additionally, patients with gynecological malignancies have an increased risk of anemia and hypoproteinemia, which are known risk factors of SSI [5,18,19,20]. A low serum albumin level may be used as an index of poor nutritional status and most commonly occurs in older, obese, and diabetes patients [13]. However, while advanced age, diabetes mellitus, ASA score, tumor entity, length of laparotomy (extension over the umbilicus), and skin closure technique showed significance for SSI in the univariable model, none were independently significant in the multivariable regression. A similar effect can be seen for FD with some risk factors (advanced age, units of crystalloids during surgery, tumor entity). These findings may be seen in contrast to other studies mentioned above, which describe risk factors but did not carry out multivariable analyzes or excluded gynecological patients [16].

Additionally, our data provide information on the incidence of FD in patients with gynecological malignancies. Compared with the data from general surgery patients, our findings showed that FD occurred more often (8.8%) in our cohort, although the results were comparable to those in older patients (10%) [2,14,16].

Table 4 lists publications that have specifically developed prediction models for SSI or FD in gynecologic malignancies.

Regarding the total number of cancer entities, FD and SSI occurred less frequently in patients with cervical cancer: typically, these patients become ill at an earlier age, are therefore less likely to have comorbidities and, in most cases, do not require bowel surgery. In comparison, due to the characteristics of the disease and its treatment, patients with ovarian cancer have a higher risk of SSI and FD: they are often older, have advanced tumor stages requiring bowel surgery, and suffer from comorbidities.

The literature regarding the context of skin closure and wound complications in gynecological cases mainly focuses on caesarean sections, with some studies reporting benefits when using subcuticular suture instead of staples, especially in obese patients [31]. Regarding longitudinal laparotomy, some studies find more wound-healing problems with staple sutures, while others demonstrate no difference [32,33]. We could not show any independent significant effect of the skin closure technique or the laparotomy length (whether extended over the umbilicus) on the rate of SSI.

### 4.3. Strengths and Weaknesses

This study has some important limitations that should be mentioned. First, the retrospective nature of this study represents an important limitation, as some potentially important variables (e.g., hypoalbuminemia) could not be included in the analysis if they were not routinely measured. Additionally, our study lacked follow-up data and information on outcomes after postoperative day 30 and, consequently, complications occurring after this timepoint are missed by this analysis. Another important limitation is the cohort size of 204 patients. Clearly, this number is too low to develop accurate prediction models and overfitting leading to exaggerated predictions may be an issue in our analysis [34]. Thus, the predictions assumed by the prediction models in this study may be inaccurate and need to be validated externally.

A notable strength of our study is the systematic investigation of interaction effects, which are usually ignored in other studies predicting wound complications (i.e., SSI and FD) [14,16,19,20]. Thus, our study provides further evidence on the interplay of various factors (e.g., length of the procedure and bowel surgery) and their contribution to the risk of wound complications.

### 4.4. Implications for Practice and Future Research

The incidence of SSI and FD is influenced by several risk factors, and the effect of one risk factor may depend on the presence of another. We suggest that operation time and bowel surgery, as well as age and BMI, show significant interaction effects. Thus, these predictors should be implemented accordingly in future prediction models for wound complications.

## 5. Conclusions

Our study identifies new aspects in the interaction between risk factors leading to SSI and FD after open surgery for gynecological cancer. The main finding of this study is the characterization of operation time as a risk factor for both SSI and FD, particularly when bowel surgery is included in the procedure. Additionally, the risk for SSI is associated with higher BMI, but this correlation is more pronounced in younger women, suggesting that, in older patients, other factors than BMI may play a critical role. In summary, we describe new aspects that can enhance the prediction of SSI and FD.

This research adds further information on the relevance of FD and SSI as complications of exploratory laparotomy in patients with gynecological malignancies. Our model may be useful for predicting risk factors for SSI and FD during the operation. We recommend paying special attention to these factors when patients undergo intestinal surgery and operation duration is prolonged. In addition, the combination of younger patients and obesity should be understood as a warning signal.

## Figures and Tables

**Figure 1 cancers-16-04157-f001:**
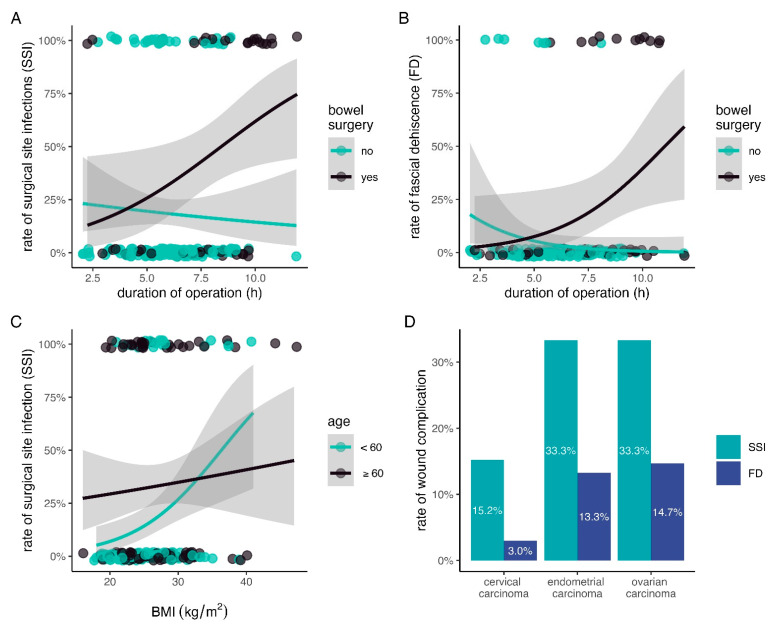
Graphical representation of statistical interactions in surgical site infection (SSI) and fascial dehiscence (FD). (**A**,**B**) The association between the duration of surgery and surgical site infection (SSI) or fascial dehiscence (FD) depends on whether bowel surgery was performed. The rate of wound complications (SSI and FD) only increased with longer operation time if bowel surgery was part of the procedure. (**C**) The rate of SSI increased with higher body mass index (BMI). However, this effect was stronger in younger patients and weaker in older patients (age was dichotomized at 60 years to illustrate this effect). The dots in (**A**–**C**) represent the individual cases in the cohort and are located at the top, if the outcome was present, and at the bottom if not (small random vertical variation was added to the dots in order to avoid overplotting and improve visualization). (**D**) Incidence of SSI and FD depending on gynecologic malignancy.

**Table 1 cancers-16-04157-t001:** Patients’ characteristics. All cases of fascial dehiscence (FD) were also counted as surgical site infections (SSI).

Characteristic	Complete Cohort	No Wound Complication	Wound Complication	
	N = 204	N = 154	SSI: N = 50	FD: N = 18
age (years)	57.5 (42.0–67.0)	54 (42.0–64.0)	65.0 (50.5–73.8)	65.5 (61.3–74.0)
BMI (kg/m^2^)	25.0 (22.0–29.0)	25.0 (22.0–29.0)	26.0 (24.0–30.8)	25.0 (23.0–27.8)
diabetes	16 (7.8%)	6 (3.9%)	10 (20.0%)	3 (16.7%)
cervical carcinoma	99 (48.5%)	84 (54.5%)	15 (30.0%)	3 (16.7%)
endometrial carcinoma	30 (14.7%)	20 (13.0%)	10 (20.0%)	4 (22.2%)
ovarian carcinoma	75 (36.8%)	50 (32.5%)	25 (50.0%)	11 (61.1%)
operation time (hours)	6.2 (5.1–7.8)	6.2 (5.0–7.4)	6.7 (5.3–8.8)	7.9 (5.5–9.8)
bowel surgery performed *	47 (23.0%)	26 (16.9%)	21 (42.0%)	11 (61.1%)
neoadjuvant chemotherapy	15 (7.4%)	10 (6.5%)	5 (10.0%)	2 (11.1%)
ASA score 1	31 (15.2%)	29 (18.8%)	2 (4.0%)	1 (5.6%)
ASA score 2	147 (72.1%)	111 (72.1%)	36 (72.0%)	14 (77.8%)
ASA score 3	24 (11.8%)	13 (8.4%)	11 (22.0%)	3 (16.7%)
ASA score 4	2 (1.0%)	1 (0.6%)	1 (2.0%)	0 (0.0%)
active smoker	47 (23.0%)	36 (23.4%)	11 (22.0%)	2 (11.1%)
hemoglobin (mmol/L) before surgery	8.1 (7.4–8.5)	8.1 (7.5–8.5)	8.1 (7.0–8.5)	8.2 (7.8–8.9)
serum protein (g/L) before surgery	72.0 (68.0–74.0)	72.0 (68.0–74.0)	72.0 (67.0–75.0)	70.0 (66.3–74.0)
units of crystalloids given during surgery	6 (5–9)	6 (5–8)	7 (5–10)	10 (5–12)
laparotomy extended over umbilicus	152 (74.9%)	109 (71.2%)	43 (86.0%)	15 (83.3%)
skin closure with subcuticular suture	48 (24.4%)	43 (29.1%)	5 (10.2%)	2 (11.8%)
skin closure with stapler	149 (75.6%)	105 (70.9%)	44 (89.8%)	15 (88.2%)

n (%); median (Interquartile range = IQR). ASA—classification of physical status (American Society of Anesthesiologist); BMI—body mass index; FD—fascial dehiscence, wound dehiscence, burst abdomen; SSI—surgical site infection. * Bowel surgery included appendectomy, larger bowel surgery like pelvic exenteration and formation of stoma—all three subgroups had a significant effect on SSI and FD and were therefore clustered into one variable.

**Table 2 cancers-16-04157-t002:** Multivariable logistic regression models for SSI prediction.

Characteristic	Univariable Regression			Multivariable Regression		
	OR	95% CI	*p*-Value	OR	95% CI	*p*-Value
operation time (per hours)	1.21	(1.03–1.42)	0.024	0.98	(0.74–1.28)	0.88
bowel surgery performed *	3.56	(1.76–7.23)	<0.001	4.59	(1.24–18.2)	0.024
operation time (per hour, if bowel surgery performed) ^‡^	1.46	(1.01–2.12)	0.044	1.52	(1.03–2.29)	0.039
age (decades)	1.44	(1.15–1.84)	0.002	1.19	(0.85–1.69)	0.31
BMI (per 10 kg/m^2^)	2.39	(1.32–4.45)	0.005	2.48	(1.12–5.70)	0.028
BMI (per 10 kg/m^2^ and per decade) ^‡^	0.59	(0.36–0.94)	0.027	0.48	(0.26–0.83)	0.013
diabetes	6.17	(2.16–19.1)	<0.001	1.94	(0.49–7.83)	0.34
laparotomy extended over umbilicus	2.48	(1.09–6.40)	0.041	0.78	(0.24–2.65)	0.69
hemoglobin (mmol/L) before surgery	0.79	(0.57–1.11)	0.17	0.89	(0.59–1.35)	0.58
units of crystalloids given during surgery	1.07	(0.97–1.18)	0.15	0.87	(0.72–1.03)	0.10
**ASA score**						
1	ref.	ref.		ref.	ref.	
2	4.70	(1.32–30.0)	0.041	2.46	(0.59–16.9)	0.27
3	12.3	(2.80–87.0)	0.003	4.26	(0.64–39.3)	0.15
4	14.5	(0.46–489)	0.093	7.11	(0.19–285)	0.25
**type of cancer**						
cervical carcinoma	ref.	ref.		ref.	ref.	
endometrial carcinoma	2.80	(1.08–7.14)	0.031	1.57	(0.46–5.20)	0.46
ovarian carcinoma	2.80	(1.36–5.92)	0.006	0.68	(0.17–2.49)	0.57
**skin closure technique**						
subcuticular suture	ref.	ref.		ref.	ref.	
stapler	3.60	(1.45–10.9)	0.011	3.05	(0.97–11.2)	0.069

* Bowel surgery included appendectomy, larger bowel surgery like pelvic exenteration and formation of stoma—all three subgroups had a significant effect on SSI and FD and were therefore clustered into one variable. ^‡^ Statistical interaction term. ASA—classification of physical status (American Society of Anesthesiologist); BMI—body mass index; OR—odds ratio; ref.—reference subgroup; SSI—surgical site infection; 95% CI—95% confidence intervals.

**Table 3 cancers-16-04157-t003:** Multivariable logistic regression models for fascial dehiscence prediction.

Characteristic	Univariable Regression			Multivariable Regression		
	OR	95% CI	*p*-Value	OR	95% CI	*p*-Value
operation time (per hour)	1.31	(1.03–1.68)	0.03	0.6	(0.34–0.98)	0.06
bowel surgery performed *	6.55	(2.41–18.9)	<0.001	8.57	(0.91–94.9)	0.06
operation time (per hour, if bowel surgery performed) ^‡^	2.38	(1.28–4.43)	0.006	2.48	(1.34–5.1)	0.007
age (per decade)	1.61	(1.13–2.42)	0.01	1.51	(0.90–2.65)	0.13
diabetes	2.66	(0.57–9.44)	0.16	1.08	(0.17–5.71)	0.93
units of crystalloids given during surgery	1.18	(1.04–1.33)	0.01	1.06	(0.84–1.32)	0.62
**ASA score**						
1	ref.	ref.		ref.	ref.	
2	3.16	(0.6–58.3)	0.28	0.49	(0.05–10.7)	0.56
3	4.29	(0.51–90)	0.22	0.31	(0.02–9.33)	0.44
4	n/a	n/a		n/a	n/a	
**type of cancer**						
cervical carcinoma	ref.	ref.		ref.	ref.	
endometrial carcinoma	4.92	(1.03–26.3)	0.05	4.01	(0.58–32.5)	0.16
ovarian carcinoma	5.50	(1.64–25.0)	0.01	0.57	(0.06–4.98)	0.61
**skin closure technique**						
subcuticular suture	ref.	ref.		ref.	ref.	
stapler	2.57	(0.69–16.7)	0.22	1.4	(0.28–10.6)	0.71

* Bowel surgery included appendectomy, larger bowel surgery like pelvic exenteration and formation of stoma—all three subgroups had a significant effect on SSI and FD and were therefore clustered into one variable. ^‡^ Statistical interaction term. ASA—classification of physical status (American Society of Anesthesiologist); OR—odds ratio; ref.—reference subgroup; SSI—surgical site infection; 95% CI—95% confidence intervals; n/a—not applicable (due to limited data in the subgroup).

**Table 4 cancers-16-04157-t004:** Summary of previous publications focusing on prediction models of SSI and FD in gynecological malignancies.

Publication (Year)	SSI (In %)	FD (In %)	Number of Patients	Significant Risk Factors
Nugent et al. (2011) [19]	34	n/a	373	BMI, pulmonary disease, albumin, prior abdominal surgery, length of surgery, pelvic drain placement, lysis of adhesions performed
Tran et al. (2015) * [17]	10.8	n/a	888	increasing BMI, increasing operative time, advanced tumor stage
Shi et al. (2021) ^‡^ [18]	14.47	n/a	318	FIGO stage IV, open surgery, duration of drainages ≥ 7 d, postoperative serum albumin < 30 g/L, postoperative blood sugar ≥ 10 mmol/L
Our publication (2024)	24.5	8.8	204	bowel surgery, operation time (if bowel surgery was performed), BMI (depending on the age of the patient)

* Study only included ovarian cancer. ^‡^ Study only included endometrial cancer. BMI—body mass index; FIGO—Fédération Internationale de Gynécologie et d’Obstétrique.

## Data Availability

The data presented in this study are available upon reasonable request. The data are not publicly available due to privacy and ethical reasons.

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
