# Peer review of "Risk Factors for Surgical Wound Infection and Fascial Dehiscence After Open Gynecologic Oncologic Surgery: A Retrospective Cohort Study"

_cancers, 2024, doi:10.3390/cancers16244157_

Round 1
Reviewer 1 Report
Comments and Suggestions for Authors
1.With only 204 patients, the study may lack the power to cover all potential variables and avoid overfitting. Study should consider a larger sample size or multicenter collaborations to enhance model generalizability.
2.As a retrospective study, data completeness depends on previous medical records, making it challenging to include certain crucial risk factors (e.g., serum albumin). Prospective studies could provide more comprehensive and consistent data.
3.The study only observes complications within the first 30 days post-surgery, potentially missing longer-term SSI and FD developments. Extending the follow-up period could offer a more comprehensive understanding of postoperative complications and their long-term impacts.
4. Some studies similar to this one, for example, Nugent et al. (2011) focused on patients undergoing gynecologic cancer surgeries, analyzing the effects of BMI, comorbidities, nutritional status, and surgical type on SSI and FD. It developed an SSI risk prediction model specifically for gynecologic oncology patients and recommended special preventive measures for high-risk individuals, which also investigate risk factors and predictive models related to surgical site infection (SSI) and fascial dehiscence (FD) in gynecologic oncology surgeries. Unfortunately, this study does not present any particularly novel findings.
Author Response
Dear reviewer,
Thank you very much for taking the time to read through this manuscript and add your precious comments, as we know that reviewing happens on a scientist’s free time. Still, this effort is essential to present valuable and consistent research. We implemented your suggestions in the manuscript and answered to all comments, hopefully to your satisfaction.
We would appreciate it, if you would take the time to reevaluate the modified manuscript and thank you in advance for your efforts.
On behalf of the authors, Carolin Hagedorn
Comment 1. With only 204 patients, the study may lack the power to cover all potential variables and avoid overfitting. Study should consider a larger sample size or multicenter collaborations to enhance model generalizability.
Comment 2. As a retrospective study, data completeness depends on previous medical records, making it challenging to include certain crucial risk factors (e.g., serum albumin). Prospective studies could provide more comprehensive and consistent data.
Response 1 and 2:
You are absolutely right, the number of patients is small and multivariable regression would lead to overfitting and thus “exaggerating” the adjusted odds ratios. We expanded our explanations on this fact in the limitation section. As a novel finding, we systematically search for statistical interactions (i.e. if the effect of certain predictors depends on other variables) which should be implemented in future prediction models. In fact, most prediction models in the area of research do not account for interactions between predictors.
Nevertheless, we clarified these points and hope these limitations are now sufficiently brought across.
Comment 3: The study only observes complications within the first 30 days post-surgery, potentially missing longer-term SSI and FD developments. Extending the follow-up period could offer a more comprehensive understanding of postoperative complications and their long-term impacts.
Response 3:
By using a 30-day post-surgery interval, we followed the definition of the CDC (Centers for Disease Control and Prevention) who chose a 30-day period
In addition, this time frame is used by various other studies as well i.e. Desale (2016); Tran (2015); Pellegrini (2017); Schiavone (2017); AlHilli (2015); vanRamshorst (2009); Petca (2022); Guo (2020); Bruce (2018); Lachiewicz (2015); Helgeland (2019); Shi (2021); Cheng (2017).
However, we agree with the reviewer that complications occurring after 30 days will be missed by this analysis and thus we added this fact into the limitations of the study.
Comment 4: Some studies similar to this one, for example, Nugent et al. (2011) focused on patients undergoing gynecologic cancer surgeries, analyzing the effects of BMI, comorbidities, nutritional status, and surgical type on SSI and FD. It developed an SSI risk prediction model specifically for gynecologic oncology patients and recommended special preventive measures for high-risk individuals, which also investigate risk factors and predictive models related to surgical site infection (SSI) and fascial dehiscence (FD) in gynecologic oncology surgeries. Unfortunately, this study does not present any particularly novel findings.
Response 4:
Thank you very much for pointing out this publication. Data, such as those from the publication mentioned, were the basis for our assumptions and study design (i.e. defining our risk factors like BMI, age, diabetes, ASA score etc.). However, the study by Nugent et al. does not account for interaction effects. As an example, this study points out that the variable “length of surgery” has a significant effect on the rate of wound complications. However, our data suggests that the effect of “length of surgery” depends on whether bowel surgery was performed (i.e. additional bacterial contamination). Consequently, we suggest that such “effect modifications” are taken into account in future prediction models, as we think that the model of Nugent et al. does overestimate the risk of wound complication in patients with long operations but without bowel surgery.
The novel finding of this study is that we explore for such interaction effects with an automated method, which systematically analyses all possible combination of variables (genetic algorithm of the glmulti-package in R). These results could help future studies to develop more accurate prediction models.
Reviewer 2 Report
Comments and Suggestions for Authors
A well written article with clearly presented results. relevant conclusions. I would like a table for discussions with comparative results of this study with other studies that confirm the results and that possibly have a larger number of cases.
Author Response
Dear reviewer,
Thank you very much for taking the time to read through this manuscript and add your precious comments, as we know that reviewing happens on a scientist’s free time. Still, this effort is essential to present valuable and consistent research. We implemented your suggestions in the manuscript and answered to all comments, hopefully to your satisfaction.
We would appreciate it, if you would take the time to reevaluate the modified manuscript and thank you in advance for your efforts.
On behalf of the authors, Carolin Hagedorn
Comment 1: A well written article with clearly presented results. relevant conclusions. I would like a table for discussions with comparative results of this study with other studies that confirm the results and that possibly have a larger number of cases.
Response 1:
Thank you very much for the positive feedback and the great idea of adding a comparison. We included a table in our manuscript that shows publications specifically developing prediction models for SSI or FD in gynecological malignancies.
Table 4. Summary of previous studies focusing on prediction models of SSI and FD in gynecological malignancies.
|
Publication (year) |
SSI (in %) |
FD (in %) |
Number of patients |
Significant risk factors |
|
Nugent et al. (2011) |
34 |
n/a |
373 |
BMI, pulmonary disease, albumin, prior abdominal surgery, length of surgery, pelvic drain placement, lysis of adhesions performed |
|
Tran et al. (2015) * |
10.8 |
n/a |
888 |
increasing BMI, increasing operative time, advanced tumor stage |
|
Shi et al. (2021) |
14.47 |
n/a |
318 |
FIGO stage IV, open surgery, duration of drainages ≥7d, postoperative serum albumin <30 g/L, postoperative blood sugar ≥10 mmol/l |
|
Our publication (2024) |
24.5 |
8.8 |
204 |
bowel surgery, operation time (if bowel surgery was performed), BMI (depending on the age of the patient) |
* Study only included ovarian cancer
‡ Study only included endometrial cancer
BMI — body mass index; FIGO –Fédération Internationale de Gynécologie et d'Obstétrique
Table 4: This table shows studies specifically developing prediction models for SSI or FD in gynecological malignancies and compares rates of SSI, FD, number of patients as well as significant risk factors.
Reviewer 3 Report
Comments and Suggestions for Authors
Some points of criticism include:
Abstarct
for a malignant gynecologic diseases (Abstarct, line 28)
Materials and Methods
In the methods, the authors state that in this cohort-study were included all patients with gynecological cancer (lines 68-69) but latter they state that patients with a negative for malignancy histology were excluded (lines 76-77). This is somewhere contradictive and should be clarified.
In their methods and results, the authors evaluated concomitant bowel surgery as a variable for SSI and FD (lines 88-89). Do they mean large bowel surgery, either intended or accidental, or small bowel? This is crucial for the development of SSI and needs to be clarified.
Were there any cultures taken from the infected wound? How did they document SSI?
Please clarify what type of sutures was used for abdominal fascia closure (line 100).
I wonder why the authors have chosen a p-value of <0.25 for variable inclusion in the multivariable regression analysis (lines 108-109).
Results
“SSI occurred in 50 (24.5%) and FD in 18 (8.8%) patients” (line 131). I assume that the 18 FD patients were part of those 50 patients with SSI. Please clarify.
Author Response
Dear reviewer,
Thank you very much for taking the time to read through this manuscript and add your precious comments, as we know that reviewing happens on a scientist’s free time. Still, this effort is essential to present valuable and consistent research. We implemented your suggestions in the manuscript and answered to all comments, hopefully to your satisfaction.
We would appreciate it, if you would take the time to reevaluate the modified manuscript and thank you in advance for your efforts.
On behalf of the authors, Carolin Hagedorn
Comment 1: Abstarct: for a malignant gynecologic diseases (Abstarct, line 28)
Response 1:
We improved the language and the sentence was changed to “A total of 204 women underwent open surgery for malignant gynecological diseases at our institution”
Comment 2: Materials and Methods: In the methods, the authors state that in this cohort-study were included all patients with gynecological cancer (lines 68-69) but latter they state that patients with a negative for malignancy histology were excluded (lines 76-77). This is somewhere contradictive and should be clarified.
Response 2:
Thank you very much for this comment. We screened all patients with suspected gynecological cancer and only included the patients with confirmed malignancy (total number 204) – therefore patients with benign histology were not included in the model and do not count into the 204 patients. A specification explaining this approach was added in our methods.
Comment 3: Materials and Methods: In their methods and results, the authors evaluated concomitant bowel surgery as a variable for SSI and FD (lines 88-89). Do they mean large bowel surgery, either intended or accidental, or small bowel? This is crucial for the development of SSI and needs to be clarified.
Response 3:
Every time, the bowel was opened, this was counted as “bowel surgery”. As the rate of accidental bowel opening was very low in this cohort, we did not differentiate between accidental and intended bowel surgery. Furthermore, bowel surgery encompassed large bowel surgery, stoma placement, and appendectomy. All three subgroups had significant effect on SSI (appendectomy: p=0.03; stoma placement: p= 0.01; large bowel surgery: p=0.001) and where therefore clustered into one variable (i.e. bowel surgery). An additional clarification was entered into our manuscript (Table 2 and 3)
Comment 4: Materials and Methods: Were there any cultures taken from the infected wound? How did they document SSI?
Response 4:
Microbiological wound swab was taken if possible but not included into the statistical analysis. We defined SSI according to the definition of the Centers for Disease Control and Prevention (CDC) including the following criterion:
- Date of event occurs within 30 days following the NHSN operative procedure AND
- involves only skin and subcutaneous tissue of the incision AND
- patient has at least one of the following
- purulent drainage from the superficial incision organism(s) identified from an aseptically-obtained specimen from the superficial incision or subcutaneous tissue by a culture or non culture based microbiologic testing method which is performed for purposes of clinical diagnosis or treatment (for example, not Active Surveillance Culture/Testing [ASC/AST])
- a superficial incision that is deliberately opened by a surgeon physician* or physician designee and culture or non-culture based testing of the superficial incision or subcutaneous tissue is not
AND
patient has at least one of the following signs or symptoms: localized pain or tenderness; localized swelling; erythema; or heat
- diagnosis of a superficial incisional SSI by a physician* or physician designee
Comment 5: Materials and Methods: Please clarify what type of sutures was used for abdominal fascia closure (line 100).
Response:
Ethicon® PDS™ II 1 loop
Comment 6: Materials and Methods: I wonder why the authors have chosen a p-value of <0.25 for variable inclusion in the multivariable regression analysis (lines 108-109).
Response:
Thank you for the question. In fact, this p-value does seem unconventional. However, it has been suggested by Hosmer and Lemeshow (see citation in the methods section) that a p-value of <0.25 is appropriate to screen for variables that should be included into multivariable analysis. It is argued that if in univariable analysis a variable has a p-value of borderline significance, it may still turn out to be statistically significant if adjusted with other variables in the multivariable analysis. Therefore, we chose a more “generous” p-value when screening for variables of interest.
Comment 7: Results: “SSI occurred in 50 (24.5%) and FD in 18 (8.8%) patients” (line 131). I assume that the 18 FD patients were part of those 50 patients with SSI. Please clarify.
Response 7:
Exactly. The Fascial dehiscences were also counted as SSI. We described this approach in the explanation of table one and specified in our manuscript.
Round 2
Reviewer 1 Report
Comments and Suggestions for Authors
The author has revised the manuscript according to the reviewers' suggestions.